# Small extracellular vesicles derived from microRNA-22-3p-overexpressing mesenchymal stem cells protect retinal ganglion cells by regulating MAPK pathway
Bo Yu [1,3], Kang Wang [2,3], Huijie Hao [1,3], Yan Liu [1], Yi Yue [1], Xiaorong Li [1], Xiaoli Xing [1] ✉ & Xiaomin Zhang [1] ✉

Glaucoma is the leading cause of irreversible blindness and is characterized by progressive retinal ganglion cell (RGC) loss and retinal nerve fiber layer thinning. Currently, no existing treatment is effective for the preservation of RGCs. MicroRNA-22-3p (miR22) and small extracellular vesicles derived from mesenchymal stem cells (MSC-sEVs) have neuroprotective effects. In this study, we apply miR22-overexpressing MSC-sEVs in an N-methyl-D-aspartic acid (NMDA)-induced RGC injury model to assess their short-term therapeutic effects and explore the underlying mechanisms. We find that mice in the miR22-sEVs-treated group have thicker retinas, fewer apoptotic cells, more reserved RGCs, better retinal function, and lower expression levels of Bax and caspase-3. MiR22-sEVs treatment promotes viability, inhibits apoptosis and inhibits Bax and caspase-3 expression in RGC-5 cells. MiR22 targets mitogen-activated protein kinase kinase kinase 12 to inhibit apoptosis by regulating the mitogen-activated protein kinase (MAPK) signaling pathway. Collectively, our results suggest that miR22-sEVs ameliorate NMDA-induced RGC injury through the inhibition of MAPK signaling pathway-mediated apoptosis, providing a potential therapy for glaucoma and other diseases that involve RGC damage.

Globally, glaucoma is the most common cause of irreversible blindness. Increased intraocular pressure (IOP), which causes retinal ganglion cell (RGC) loss, is a leading risk factor for initiation and progression of the disease[1]. Current effective therapies are limited to IOP-lowering methods, including medication, laser treatment, and surgery[2]. However, some patients experience disease progression even after ideal IOP control, indicating that additional factors are involved in glaucomatous RGC damage[3]. This includes glutamate neurotoxicity, which contributes to RGC apoptosis in glaucoma and other retinal diseases. Thus, animal models to study RGC apoptosis and its protection have been generated by intravitreal injection of N-methyl-D-aspartic acid (NMDA)[4]. NMDA receptor overactivation elicits

a rapid influx of calcium, which triggers downstream pro-apoptotic signaling cascades, leading to RGC apoptosis[5]. Based on the mechanism of neurotoxicity, neuroprotective therapeutic strategies beyond lowering IOP are required.

Small extracellular vesicles (sEVs) are lipid bilayer membrane-bound vesicles of endocytic origin, measuring less than 200 nm in diameter, that contain genetic materials such as DNA, mRNA, miRNA, and proteins. They perform a wide range of biological functions and transport their contents among cells[6]. Mesenchymal stem cell-derived sEVs (MSC-sEVs) mediate the beneficial effects of mesenchymal stem cells (MSC) in various disease models, such as myocardial damage[7], renal injury[8], and autoimmune

[1]Tianjin Key Laboratory of Retinal Functions and Diseases, Tianjin Branch of National Clinical Research Center for Ocular Disease, Eye Institute and School of Optometry, Tianjin Medical University Eye Hospital, No. 251, Fukang Road, Nankai District, Tianjin 300384, China. [2]Binzhou Medical University Hospital, No. 661, Huanghe 2nd Road, Binzhou City, Shandong 256600, China. [3]These authors contributed equally: Bo Yu, Kang Wang, Huijie Hao. ✉e-mail: xxlteh@126.com; xzhang08@tmu.edu.cn

diseases[9]. Especially in neurodegenerative diseases, MSC-sEVs show therapeutic potential because of their anti-inflammatory and neuroprotective functions inherited from the parent cells[10]. Our previous study revealed the therapeutic effects of MSC-sEVs in experimental autoimmune uveitis, retinal laser injury, and experimental retinal detachment[11]. Therefore, we believe that MSC-sEVs are suitable for ocular treatment for the following reasons. First, MSC-sEVs are safer for the eye because of their lower immunogenicity and risk of tumorigenesis. Second, because of the lipid bilayer membrane, their active components are protected from degradation when stored or transported relatively long. Finally, they can easily pass through biological barriers and do not cause vitreous opacities owing to their small size.

The 22-nucleotide miRNA, miRNA-22-3p (miR22), which plays an essential role in regulating neuronal injury and inflammation, was initially discovered in HeLa cells[12]. It is expressed in various body tissues and cell types, including the brain, neurons, and glia[13,14]. Several studies reported that miR22 exerts neuroprotective effects in cerebral ischemia[15], Alzheimer's disease (AD)[16], and Parkinson's disease (PD)[17]. In ocular diseases, miR22 was shown to play a protective role in retinal pigment epithelial damage[18]. In the current study, we investigated the role of sEVs derived from miR22-over-expressing MSC (miR22-sEVs) in animal and cell models of RGC damage. The results showed that miR22-sEVs reduced RGC apoptosis, preserved retinal structure and function, and inhibited the expression of apoptotic cytokines. Bioinformatics and dual-luciferase reporter analyses showed that *mitogen-activated protein kinase kinase kinase 12 (Map3k12)* was the target gene of miR22. Further studies showed that miR22 inhibited the protein expression of MAP3K12, ERK, p38, and caspase-3, indicating that miR22 inhibits RGC apoptosis by regulating the mitogen-activated protein kinase (MAPK) signaling pathway.

## Results

### Characterization of MSC and MSC-sEVs
MSC were characterized by the positive expression of CD90 and CD105 markers and the negative expression of CD34 and CD45 markers, as we previously reported[19]. After 48 h of transfection, green fluorescence was observed in the MiR22-overexpressing lentivirus transfected MSC (miR22-MSC) and control scramble lentivirus transfected MSC (con-MSC), indicating the success of lentivirus transfection (Fig. 1a). Then sEVs were characterized by transmission electron microscopy (TEM), NanoSight NS300 and western blotting. TEM images showed that sEVs were round, vesicle-shaped with a diameter around 100 nm (Fig. 1b). NanoSight showed that the peak size of sEVs was around 100 nm (Fig. 1c). Western blotting results indicated that sEVs were positive for CD9, CD81, and TSG101, and were negative for GM130. Cells were positive for β-actin, and GM130 (Fig. 1d). RT-PCR showed miR22-MSC demonstrated higher expression of miR22 (Fig. 1e). miR22-sEVs expressed higher levels of miR22 (Fig. 1f).

### Enhanced beneficial effect of miR22-sEVs in NMDA-induced RGC injury animal model
To examine and compare their therapeutic effects, we intravitreally injected MSC-sEVs and miR22-sEVs in a NMDA-induced RGC injury model. The intraocular pressure was normal an hour after intravitreal injection. On day 7 post-injury, HE staining was performed to observe retinal histological changes. Results showed that the cell number in the ganglion cell layer decreased in the PBS group more significantly than that of the three treatment groups after NMDA injection (Fig. 2a). TUNEL staining was performed to examine retinal apoptosis 3 days after injury. Results showed that apoptotic cells were mainly located in the ganglion cell layer. Some were located in the inner nuclear layer (Fig. 2b). Treatment with miR22-sEVs was found to significantly reduce the number of apoptotic cells compared with the PBS group. Although a trend of decreased number of apoptotic cells was observed in MSC-sEVs and con-sEVs groups, there were no statistical differences among the PBS, MSC-sEVs, and con-sEVs groups (Fig. 2c). On day 7 post-injury, we further performed OCT to measure and compare the retinal thickness of different groups. Images of OCT showed thinning of the

retinal layers after injury (Fig. 2d). The retinal thickness of the PBS group, MSC-sEVs group, and con-sEVs group were significantly decreased compared with the normal group. The retinal thickness of miR22-sEVs group was significantly increased compared with that of the PBS group. There was a tendency of increasing retinal thickness in MSC-sEVs group and con-sEVs group, but it was not statistically significant (Fig. 2e). To observe the survival of RGCs, we examined the flat-mounts of mouse retina stained with RBPMS. On day 7 post-injury, the number of RGCs of the four injury groups were fewer than that of the normal group. Treatment by MSC-sEVs, con-sEVs, or miR22-sEVs effectively promoted RGC survival, among which miR22-sEVs demonstrated the best efficacy (Fig. 2f, g).

In addition to retinal histological observation, we also performed ERG to examine the retinal function. Dark-adapted ERG revealed markedly reduced responses 7 days post-injury. B-wave amplitudes showed a significant decrease (Fig. 2h). A-wave amplitudes of the three treated groups showed a trend to decrease compared with the PBS group. B-wave amplitudes of MSC-sEVs group, con-sEVs group, and miR22-sEVs group were larger than those of the PBS group, which indicated better retinal function (Fig. 2i).

### MiR22-sEVs suppressed RGC injury-related apoptosis
Our study has indicated that intravitreal injection of miR22-sEVs could inhibit NMDA-induced RGC apoptosis, improve the layered structure of the retina, and preserve retinal function. Therefore, we examined the mRNA expressions of Bax and caspase-3, two important apoptotic factors 3 days after injury. The post-injury upregulated Bax and caspase-3 mRNA expression was significantly decreased by the treatment of miR22-sEVs (Fig. 3a). The caspase-3 expression level of the con-sEVs and MSC-sEVs groups showed a trend to decrease compared with the PBS group with no significant difference (Fig. 3b). As with the mRNA expression, Bax and caspase-3 protein expression levels followed similar patterns (Fig. 3c, d).

### MiR22-sEVs promoted RGC viability and suppressed RGC apoptosis in vitro
To verify whether the in vitro results were consistent with the results in vivo, RGC-5 cell line was used for cell model construction. CCK-8 results revealed that the viability of RGCs significantly decreased 72 h after NMDA exposure. Treatment with MSC-sEVs or con-sEVs showed enhanced viability. RGC treated with miR22-sEVs had the strongest viability (Fig. 4a). Annexin V/PI flow cytometry assay showed that treatment with different types of sEVs could suppress the RGC apoptosis induced by NMDA. The best suppressive effect was achieved by miR22-sEVs (Fig. 4b). Similarly, the protein expression of Bax and caspase-3 were upregulated after NMDA exposure (Fig. 4c). MiR22-sEVs could downregulate the protein expression of Bax significantly compared with the PBS group. Compared with the other three NMDA-treated groups, miR22-sEVs treatment inhibited the protein expression of caspase-3 significantly (Fig. 4d). To further test the apoptosis activity, we pretreated RGC-5 cell line with Z-VAD-FMK, a pan-caspase inhibitor, for 1 h before NMDA treatment. Then the protein expression level of cleaved PARP was evaluated by western blot in the miR22-sEVs group and the control group. Results showed that the expression of cleaved PARP was strikingly inhibited by Z-VAD-FMK pretreatment. MiR22-sEVs treatment downregulated the protein expression of cleaved PARP. The results demonstrated that miR22-sEVs inhibited NMDA-induced apoptosis mainly through a caspase-dependent pathway (Fig. 4e).

### MiR22 targets to 3'-UTR of *Map3k12* to suppress apoptosis
138 potential target genes of miR22 were obtained by the intersection of prediction results of TargetScan, DIANA, and miRDB databases (Fig. 5a). KEGG pathways of the above targets were enriched using DAVID. The results showed that these predicted target genes were significantly associated with endocytosis, PI3K-Akt signaling pathway, MAPK signaling pathway, and so on (Fig. 5b). After a literature search, we found that among six or more genes enriched pathways, only PI3K-Akt signaling pathway and MAPK signaling pathway have correlation records with retinal

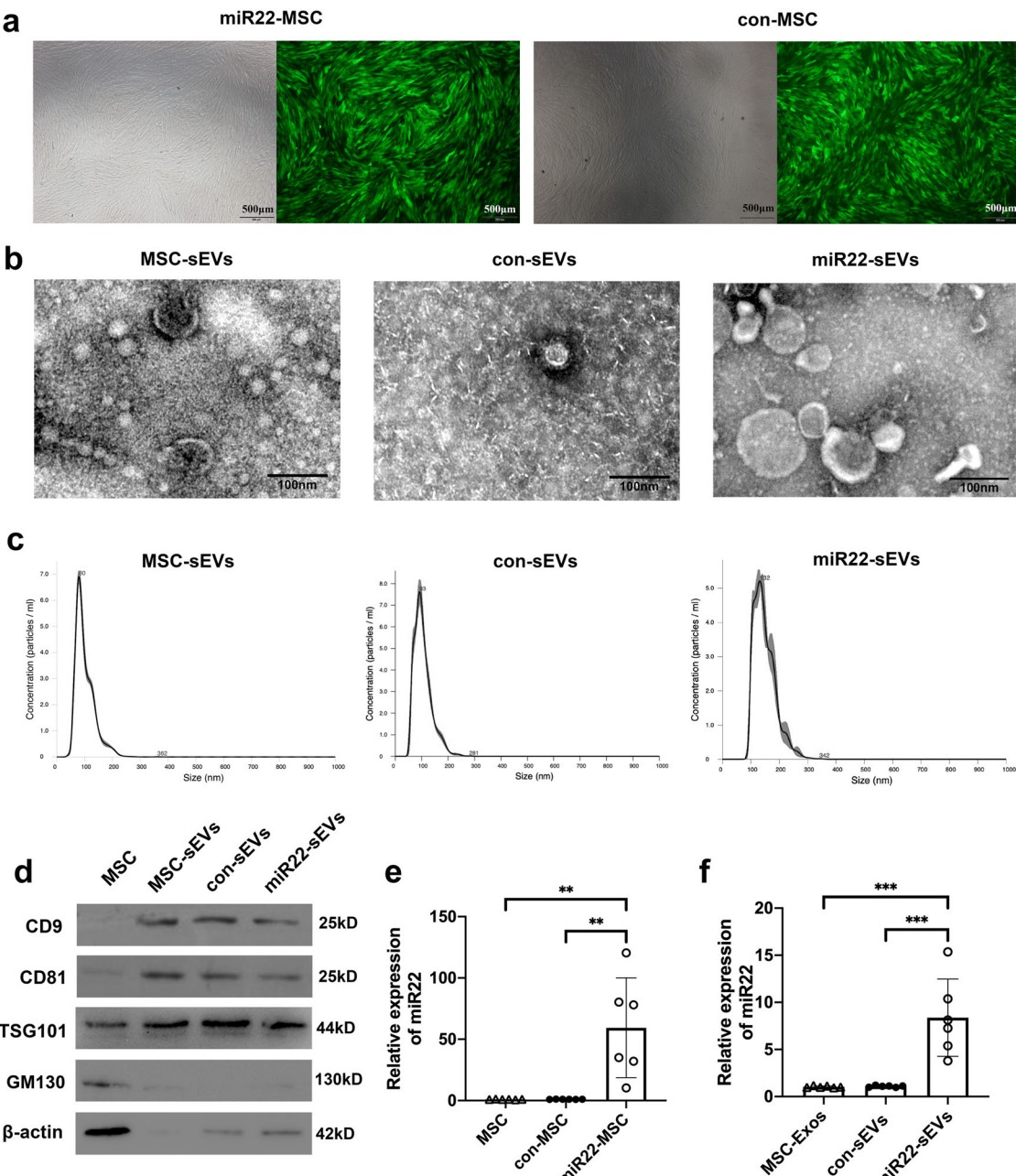

**Fig. 1 | Characterization of MSC and MSC-sEVs. a** Brightfield and darkfield light microscopy images of MSC transfected with miR22-overexpressing lentivirus (miR22-MSC) or scramble lentivirus (con-MSC). Scale bar = 500 μm.
**b** Transmission electron microscopy images of sEVs. Scale bar = 100 nm. **c** Analysis of size distribution of sEVs. **d** Western blotting for the detection of the specific markers (CD9, CD81 and TSG101) of sEVs. GM130 was used as a negative marker.

**e** Validation of the expression of miR22 in MSC, con-MSC, and miR22-MSC by qRT-PCR. Data are presented as means ± SD from 5 independent experiments.
**f** Validation of the expression of miR22 in MSC-sEVs, con-sEVs and miR22-sEVs by qRT-PCR. Data are presented as means ± SD from 5 independent experiments.
**P < 0.01, ***P < 0.005.

inflammation or apoptosis. However, it has been reported that the suppression of PI3K-Akt signaling pathway usually causes cell apoptosis, which is opposite to our results. We next examined the expression of six genes (*Max, Arrb1, Csf1r, Map3k12, Akt3, Tgfbr1*) enriched in the MAPK signaling pathway in the RGC-5 cell line after miR22 mimics treatment. After the RGC-5 cell line was transfected with miR22 mimics, the expression of miR22 was upregulated (Fig. 5c). RT-PCR results revealed that the mRNA expression level of *Max, Arrb1*, and *Csf1r* showed no differences between the control group and the miR22 mimics group. *Map3k12, Akt3*, and *Tgfbr1* mRNA expression were significantly downregulated in the miR22 mimics group. *Map3k12* showed the greatest difference in expression (Fig. 5d).

We predicted that *Map3k12* is the target gene of miR22. A dual-luciferase reporter assay was performed to verify this prediction. Results showed that miR22 could directly bind to the 3′UTR of *Map3k12* and downregulated its expression, indicating that *Map3k12* is the target of miR22 (Fig. 6a–c). To further explore the mechanism of miR22's inhibitive role in RGC apoptosis, RGC-5 cell line was transfected with miR22 mimics or miR22 inhibitors. We examined the protein expression of MAP3K12, ERK, p38 and caspase-3. Western blot results showed that miR22 mimics inhibited MAP3K12, ERK, p38, and caspase-3 protein expression. MiR22 inhibitors promoted the protein expression of MAP3K12, ERK, p38, and caspase-3 (Fig. 6d).

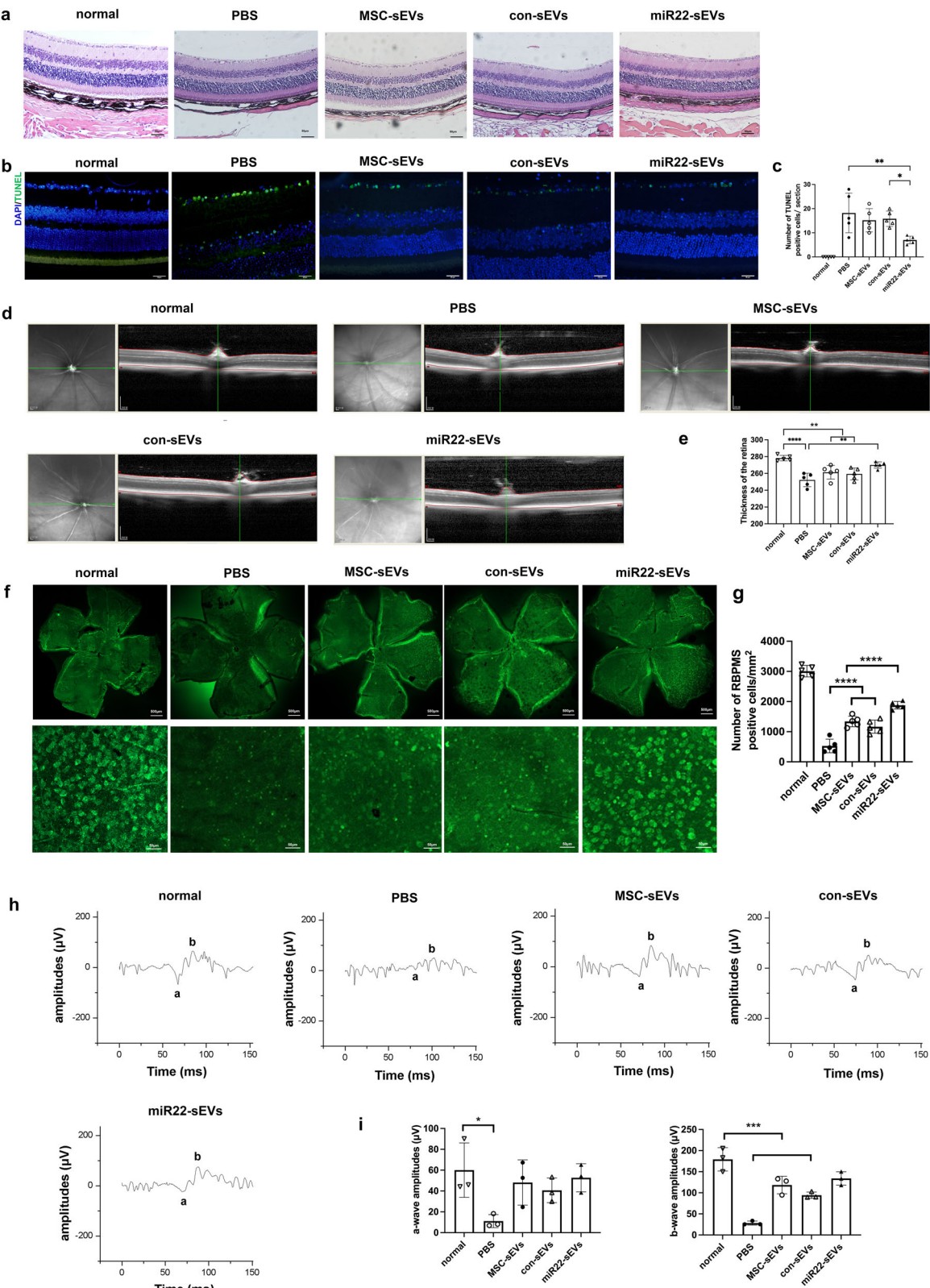

**Fig. 2 | MiR22 enhanced the beneficial effect of MSC-sEVs in NMDA-induced RGCs injury mouse model. a** Representative example of retinal HE staining images 7 days after injury. Scale bar = 50 μm. **b** Representative examples of retinal TUNEL staining images 3 days after injury. Scale bar = 50 μm. **c** TUNEL positive cells counting of each group 3 days after injury. Data are presented as means ± SD from 5 independent experiments. **d** Representative examples of OCT images 7 days after injury. **e** Retinal thickness analysis of each group 7 days after injury. Data are presented as means ± SD from 5 independent experiments. **f** Representative

examples of RGC staining retinal flat mounts images 7 days after injury. scale bar = 500 μm (up) or 50 μm (down). **g** RGCs counting of each group 7 days after injury. Data are presented as means ± SD from 5 independent experiments. **h** Representative examples of dark-adapted ERG waveforms 7 days after injury. **i** Analysis and comparison of a wave and b wave amplitudes among five groups 7 days after injury. Data are presented as means ± SD from 3 independent experiments. *P < 0.05, **P < 0.01, ***P < 0.005, ****P < 0.001.

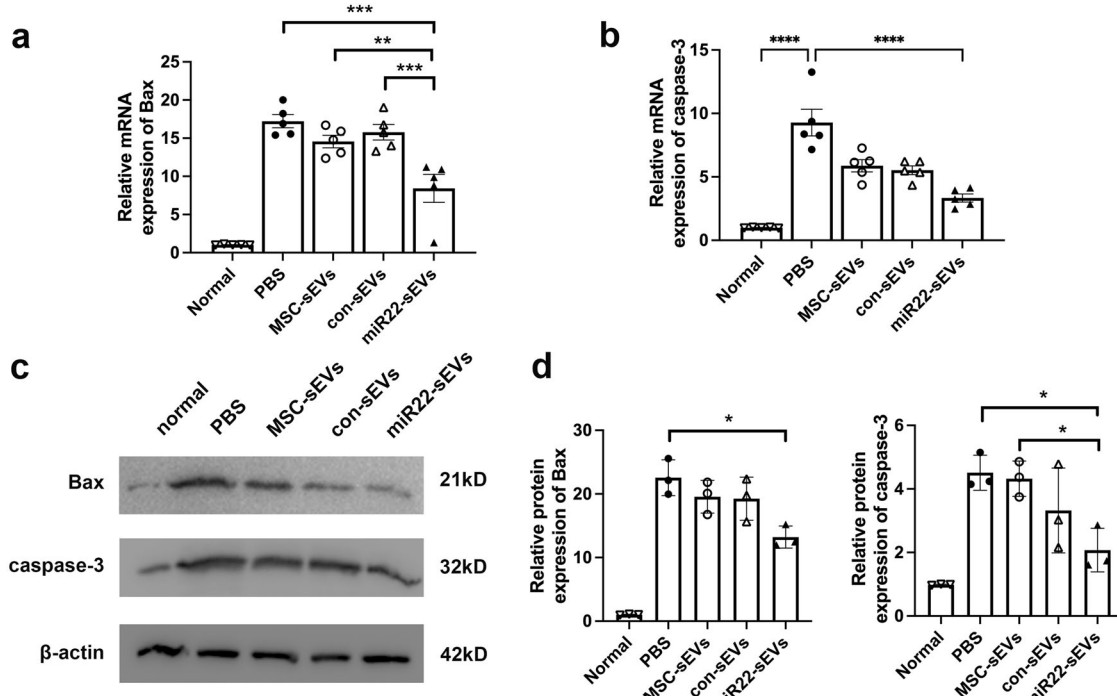

**Fig. 3 | MiR22-sEVs and MSC-sEVs suppressed the mRNA and protein expressions of Bax and caspase-3 in retina. a** Relative mRNA expression of Bax in retinas from different groups 3 days after injury. Data are presented as means ± SD from 5 independent experiments. **b** Relative mRNA expression of caspase-3 in retinas from different groups 3 days after injury. Data are presented as means ± SD from 5 independent experiments. **c** Western blotting analysis of Bax in retinas from different groups 3 days after injury. Data are presented as means ± SD from 3 independent experiments. **d** Western blotting analysis of caspase-3 in retinas from different groups 3 days after injury. Data are presented as means ± SD from 3 independent experiments. *$P < 0.05$, **$P < 0.01$, ***$P < 0.005$, ****$P < 0.001$.

## Discussion

The current study provides specific evidence of the ability of MSC-sEVs and miR22-sEVs to alleviate RGC injury. Using an NMDA-induced neurotoxicity mouse model characterized by acute RGC apoptosis and degeneration, we demonstrated that the short-term survival of RGCs and preservation of retinal function was achieved by intravitreal injection of MSC-sEVs. We also established miR22-overexpressing MSCs by lentivirus transfection and collected sEVs with high expression levels of miR22. MiR22-sEVs showed a better therapeutic effect against RGC injury than MSC-sEVs.

Rat or human MSC have been shown to be effective in the treatment of various diseases in animal model. Due to the anticipation of future clinical use, there is a growing emphasis on researching the utilization of human MSC. Several studies have demonstrated the therapeutic impact of human MSC in models of injury and degeneration in RGC. For example, one research group found that the transplantation of MSC into the vitreous cavity was protective in aged rats with RGC loss[20]. Another study found that the intravitreal injection of MSC promoted sustained neuroprotection of RGC in a rat model of optic nerve injury[21]. Conditioned media (CM) or sEVs of MSC were also shown to protect injured RGC. MSC-CM can help prevent RGC loss and attenuate apoptosis after retinal ischemia[22], while MSC-sEVs promote RGC survival in a rat optic nerve crush model[23]. The findings of these prior investigations were comparable to the current study.

In a mechanistic study, the first recognized mechanism of CM was the direct differentiation of MSC into neurons[24]. After the utilization of MSC-CM, the secretion of trophic factors is believed to contribute to its neuroprotective potential, at least in part. Various studies have shown that miRNAs encapsulated in MSC-sEVs are important functional molecules. MSC regulates neurite outgrowth via exosomal transfer of miR-133b to neural cells[25]. Mouse MSC-sEVs also attenuate myocardial ischemia-reperfusion injury in mice by shuttling miR-182, which modifies the polarization status of macrophages[26]. Finally, Human MSC-sEVs

ameliorate ischemic acute kidney injury and promote tubular repair via miR-125b-5p[8].

In addition to their role as a therapeutic agent, MSC-sEVs can serve as a tool for gene, protein, and drug delivery. Modifying MSC or MSC-sEVs can optimize their neuroprotective functions and cargo contents. SEVs derived from miR-25-overexpressing rat MSC possess enhanced neuroprotection in a rat spinal cord ischemia model[27]. MiR-146a-5p-enriched rat MSC-sEVs offer functional improvements by inhibiting inflammation and apoptosis in a rat intracerebral hemorrhage model[28]. TNF-α stimulation could enhance the neuroprotective effect of human MSC-sEVs in a mouse model of retinal ischemia-reperfusion injury[29]. SEVs derived from ginkgolide A-pretreated human MSC significantly rescued 6-hydroxydopamine-induced cell death in an in vitro model of PD[30], while those derived from pigment epithelium-derived factor-modified human MSC ameliorated cerebral ischemia-reperfusion injury in vitro and in vivo[31].

In the current study, we modified MSC by transfecting them with a lentivirus carrying miR22 and then collected their sEVs. MiR22 has been shown to be neuroprotective in neurological disorders through antioxidant and anti-inflammatory mechanisms[12]. Since, miR22 could be easily degraded by enzymes in body fluids[32]. Therefore, the current design did not include the treatment group with miR22 alone. SEVs can be used for RNAs or miRNAs encapsulation to protect them from degradation by RNases[33]. The combination of miR22 and MSC-sEVs enhanced each other's neuroprotective effects and prolonged their action time. Our results indicated that miR22 could enhance the beneficial effects of MSC-sEVs in treating NMDA-induced RGCs injury.

We also examined the gene and protein expression levels of the pro-apoptotic enzyme, Bax and caspase-3. The expression level of Bax and caspase-3 in the miR22-sEVs group was significantly lower than those in the other treatment groups. These results revealed the anti-apoptotic function of miR22-sEVs. The result is consistent with previous studies demonstrating the protective effects of miR22 overexpression via its anti-apoptotic effects[34]. This includes the hippocampus of patients with AD where the expression of

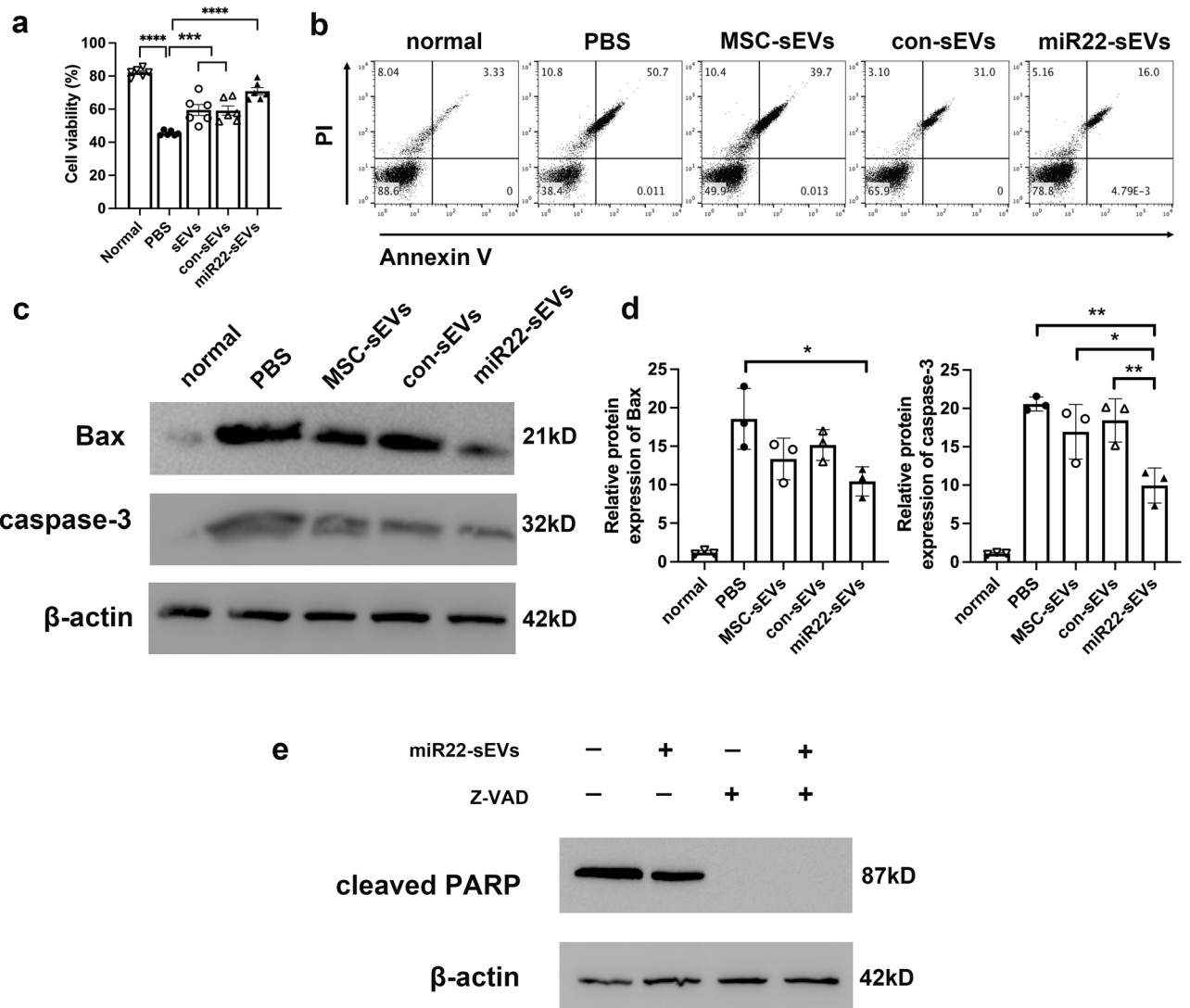

**Fig. 4 | MiR22-sEVs promoted RGC viability and suppressed RGC apoptosis in NMDA-induced RGC injury in vitro. a** CCK-8 analysis of cell viability in each group. Data are presented as means ± SD from 5 independent experiments. **b** Annexin V/PI flow cytometry analysis of RGC apoptosis in each group. **c** Western blotting analysis of Bax protein expression in each group. **d** Western blotting analysis of caspase-3 protein expression in each group. Data are presented as means ± SD from 3 independent experiments. **e** Western blotting analysis of cleaved PARP protein expression in NMDA-induced RGC-5 injury model with or without Z-VAD-FMK pretreatment. *$P < 0.05$, **$P < 0.01$, ***$P < 0.005$, ****$P < 0.001$.

miR22 improved learning and memory dysfunction by inhibiting neuronal apoptosis[35]. Furthermore, in starvation-induced cardiac injury, miR22 is potentially therapeutic by inhibiting apoptosis[36]. According to the specific circumstances in which a miRNA is expressed, each miRNA typically targets different genes, and different miRNAs can target the same gene. Similarly, miR22 plays multiple functional roles by targeting different genes through various biological signaling pathways[16,37].

MAPK signaling pathways regulate a variety of biological processes, such as proliferation, differentiation, survival, and death. The abnormal activation of MAPK signaling pathways contributes to neuroinflammatory responses and neuronal death, which are associated with the pathogenesis of AD and PD[38]. A KEGG pathway enrichment analysis showed that miR22 targets genes that were mainly enriched in the MAPK signaling pathway. Our subsequent experiments verified that miR22 protected RGCs by regulating the MAPK signaling pathway, which is consistent with previous studies. For example, one study highlighted that miR22 could repress osteoblast viability by inactivating p38 MAPK/c-Jun N-terminal kinase (JNK) signaling[39]. In a rat myocardial ischemia-reperfusion model, the overexpression of miR22 exerted a protective effect by suppressing the p38

MAPK-related signaling pathway[40]. The exposure of cells to stress activates JNK and p38 MAPK. MAPK-related signaling pathways play key roles in cell survival, apoptosis, and death. Extensive research has suggested that these kinases function in different ways to integrate signals at different transmission points, which would regulate caspase activation and affect apoptosis[41]. The current study found that, after transfection with miR22 mimics, the protein expression levels of MAP3K12 and the downstream expression of ERK, p38, and caspase-3 were decreased. These results indicate that miR22 protects RGCs from apoptosis by inhibiting the MAPK signaling pathway.

The present study had several limitations. First, we only studied the short-term therapeutic effects of MSC-sEVs and miR22-sEVs. Additional research is required to evaluate the extended duration necessary for observing the treatment's long-term effectiveness. Second, the scope of our mechanism study was exclusively concentrated on miR22; therefore, in future studies, we will thoroughly investigate the mechanism of MSC-sEVs, enhancing the research's comprehensiveness. Third, further experiments should be performed to investigate whether other target genes of miR22 are related to its protective effect against NMDA-induced RGC injury.

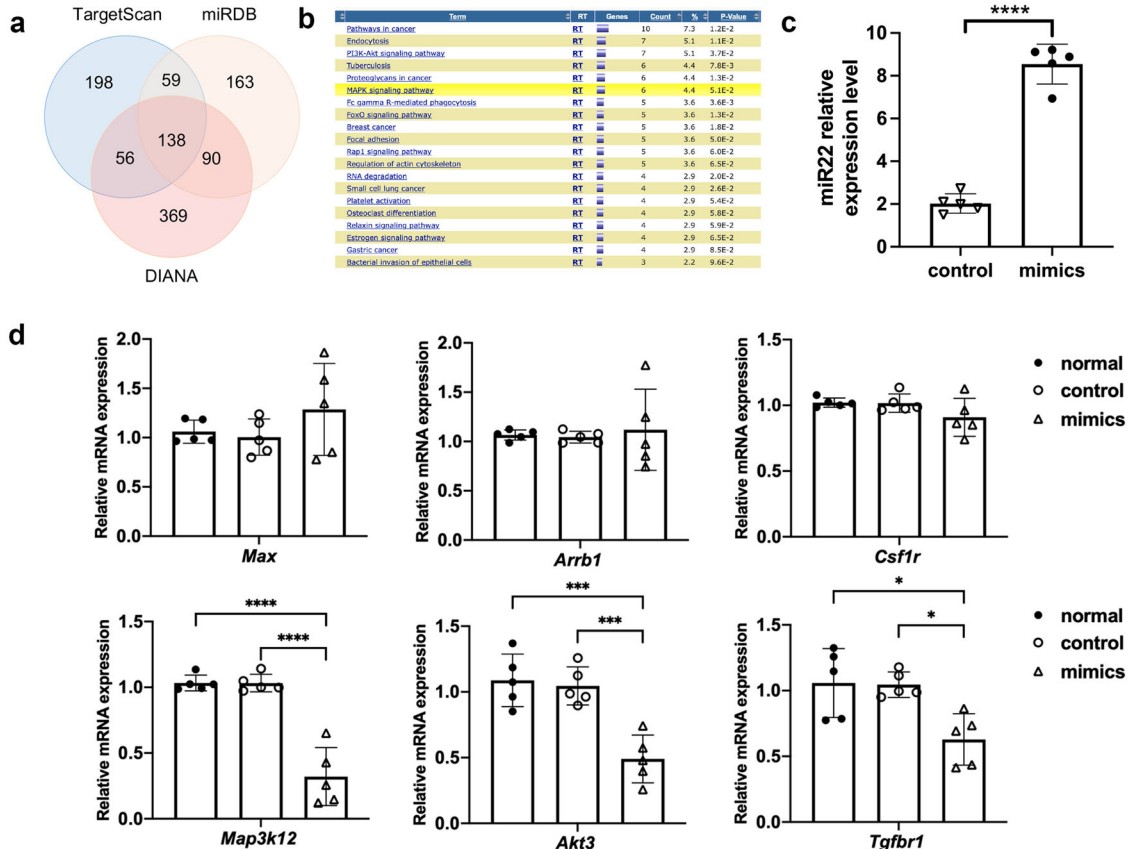

**Fig. 5 | Target gene prediction and verification of miR22. a** Predicted target genes of miR22 by TargetScan, DIANA and miRDB databases. **b** KEGG pathway analysis of the candidate target genes of miR22. **c** MiR22 expression in RGC-5 cell line post transfected with miR22 mimics. Data are presented as means ± SD from 5 independent experiments. **d** MRNA expression level of *Max, Arrb1, Csf1r, Map3k12, Akt3, Tgfbr1* in RGC-5 cell line post transfected with miR22 mimics. Data are presented as means ± SD from 5 independent experiments. *$P < 0.05$, ***$P < 0.005$, ****$P < 0.001$.

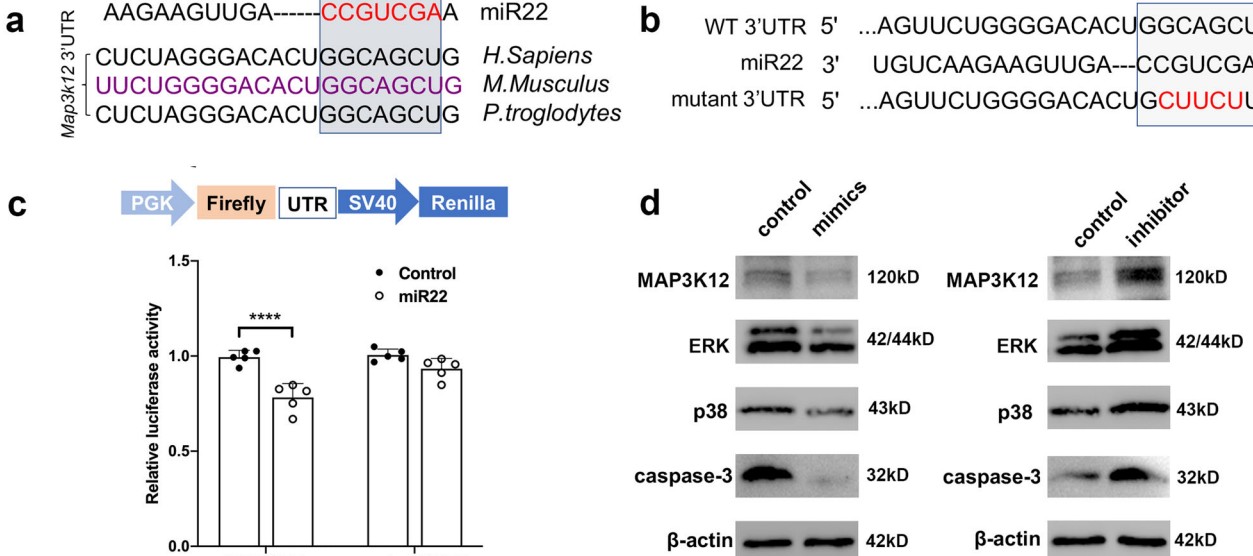

**Fig. 6 | MiR22 targets to 3'UTR of *Map3k12* and effects on MAPK pathway. a** The seed sequence of miR22 is complementary to *Map3k12* 3'UTR and is highly conserved among different species. **b** Binding site of *Map3k12* 3'UTR (WT-3'UTR) with miR22 and designed *Map3k12* 3'UTR mutant sequence (Mutant-3'UTR). **c** miR22 reduced the luciferase activity of the vector with *Map3k12* 3'UTR, but the mutant version abrogated the repressive ability. Data are presented as means ± SD from 5 independent experiments. **d** Western blotting presentation of MAP3K12, ERK, p38, and caspase-3 in RGC-5 cell line post transfected with miR22 mimics or miR22 inhibitors. ***$P < 0.005$.

In summary, MSC-sEVs exert short-term protective effects against NMDA-induced RGC injury. Genetic modification of MSC-sEVs to contain elevated miR22 levels could enhance their neuroprotective effects by targeting MAP3K12 to inhibit MAPK signaling pathway-mediated apoptosis. Thus, MSC-sEVs with high expression of miR22 might represent a novel approach for treating nerve cell damage-related diseases.

## Methods

### Animals
Female C57BL/6 mice (6–8 weeks of age) purchased from Vital River Experimental Animal Center (Beijing, China) were enrolled in the study. Mice were maintained in specific pathogen-free (SPF) conditions (temperatures of 21 ~ 25 °C with 45 ~ 65% humidity) on a 12 h dark/light cycle with free access to food and water. All animal experiments comply with the ARRIVE guidelines. All experimental procedures were in accordance with the statement for the use of animals in ophthalmic and vision research of the Association for Research in Vision and Ophthalmology (ARVO) and approved by the Laboratory Animal Care and Use Committee of Tianjin Medical University Eye Hospital (No. TJYY20201221035). Mice were randomly divided into five groups (the control group, the PBS group, the MSC-sEVs group, the con-sEVs group, and the miR22-sEVs group) using a random number table.

### Culture of MSC
Human umbilical cord-derived MSC were offered by Beilai Biological Co., Ltd. (Beijing, China). MSC were cultured and identified as follows[42]. Human umbilical cords were washed and cut into pieces in PBS, and then were sequentially digested with collagenase II and trypsinization at 37 °C. Dissociated cells were collected and were cultured in a complete medium (90% Dulbecco's modified Eagle's medium (DMEM)/F12, 10% fetal bovine serum (FBS), 100 U/mL penicillin, and 100 μg/mL streptomycin) at 37 °C supplied with 5% $CO_2$. MSC were tested for CD34, CD45, CD90 and CD105 expression by flow cytometry.

### Lentivirus transfection
MiR22-overexpressing lentivirus and control scramble lentivirus were purchased from Hanheng Biotechnology Co., Ltd (Shanghai, China). MSC at passage 2 were grown to 60% confluency prior to lentiviral transfection at the multiplicity of infection (MOI) of 50. All transductions were performed with polybrene at the concentration of 8 μg/mL for 8 h. After 48 h, green fluorescence was observed under a fluorescent microscope.

### Quantitative real-time RT-PCR (qRT-PCR) of miR22
Total RNA of MSC (normal MSC, miR22-overexpressing MSC (miR22-MSC) and control scramble lentivirus transfected MSC (con-MSC)) and sEVs (MSC-sEVs, sEVs derived from miR22-MSC (miR22-sEVs) and sEVs derived from con-MSC (con-sEVs)) were extracted using a universal RNA purification kit (EZB-RN4, EZBioscience, USA). Then miRNAs were reversed into cDNA using a miRNA reverse transcription kit (EZB-miRT4, EZBioscience, USA) according to the manufacturer's protocol. RT-PCR was carried out in 384-well plates containing primers, cDNA and EZB-miProbe-R1 (EZBioscience, USA) by ABI 7900 fast (Applied Biosystems, USA). MiRNA relative expression levels were calculated by the $2^{-\Delta\Delta Ct}$ method, with U6 as the internal control. Primers of U6 and miR22 are shown in Supplementary Table 1.

### MSC-sEVs collection and identification
MSC at passage 3–5 were cultured in FBS-free conditioned medium. The cultured supernatants were collected 48 h after passaging and subjected to gradient centrifugation at the speed of $200 \times g$ for 10 min, $2000 \times g$ for 20 min, $10000 \times g$ for 30 min and $110,000 \times g$ for 70 mins twice[19]. Precipitates were resuspended in PBS for the last centrifugation step at the speed of $110,000 \times g$ for 70 min at 4 °C. SEVs were resuspended to the protein concentration of 0.5 mg/mL in PBS. The identification of sEVs was performed[43]. The morphology of sEVs was visualized under a Phillips CM10

electron microscope (Phillips Electron Optics, Eindhoven, The Netherlands). 10 μl sEVs suspension was placed onto Formvar-carbon coated copper grids. The grid was then moved to a solution of phosphotungstic acid (50 μl, pH 7.0) for 5 min. After air-drying, the sample was examined under an electron microscope at 80 kV. The size distribution of sEVs was measured using NanoSight NS300 (Malvern, UK). CD9, CD81, TSG101, GM130 and β-actin were detected using western blotting (anti-CD9 antibody, ab236630, 1:1000, abcam, USA; anti-CD81 antibody, ab109201, 1:1000, abcam, USA; anti-TSG101 antibody, 28283-1-AP, 1:2000, proteintech, China; anti-GM130 antibody, ab52649, 1:1000, abcam, USA; anti-β-actin antibody, 4970 S, 1:1000, Cell Signaling Technology, USA).

### Induction and treatment of the animal model
Mice were anesthetized by intraperitoneal injection of xylazine hydrochloride at 10 mg/kg body weight and zoletil at 65 mg/kg body weight. 0.5% tropicamide Phenylephrine eye drops (Santen) were used for mydriasis. For the PBS group, 1 μL NMDA (M3262, Sigma-Aldrich, USA) (20 mM) + 1 μL PBS was intravitreally injected into the right eye of each mouse. For the other three treatment groups, 1 μL NMDA (20 mM) + 1 μL MSC-sEVs (0.5 mg/mL), 1 μL NMDA (20 mM) + 1 μL con-sEVs (0.5 mg/mL) or 1 μL NMDA (20 mM) + 1 μL miR22-sEVs (0.5 mg/mL) was intravitreally injected to the right eyes of mice in each group individually. IOP was measured an hour later to ensure the security of intravitreal injection.

### TUNEL staining
Eyes were collected, fixed, dehydrated and embedded in paraffin at 3 days post-treatment. 4 μm-thick coronal sections of the retinas were prepared for TUNEL staining. Paraffin-embedded sections were then deparaffinized. The TUNEL assay kit (11684817910, Roche, Germany) was used to detect apoptosis according to the manufacturer's protocol. The number of apoptotic cells per section were counted and analyzed.

### OCT examination
Retinal structures were visualized at 7 days post-treatment by OCT (Heidelberg Engineering, Heidelberg, Germany). Pupils were dilated with 0.5% tropicamide Phenylephrine eye drops. Artificial tears were used to maintain corneal clarity. Linear scans were performed to obtain high-resolution images. The OCT system software was used to measure retinal thickness at a distance of 1500μm away from the optic nerve.

### Immunofluorescence labeling of retina flat-mounts
Eyes were enucleated at 7 days post-treatment. Enucleated eyes were immersed in freshly prepared 4% paraformaldehyde for 25 min. Corneas and lens were removed, and retinas were segmented into four flaps. Retina flat-mounts were washed with PBS containing 0.5% Triton X-100. Nonspecific binding was prevented by blocking with 10% goat serum for 1 h, and primary antibody RBPMS (ABN1362, 1:200, Merck Millipore, Germany) was applied overnight at 4 °C. Then, the secondary antibody (Alexa Fluor 488 goat anti-rabbit, ab150077, 1:200, abcam, USA) was incubated for 2 h at room temperature. Flat-mounts were visualized under a confocal microscope. RGCs were counted by ImageJ software.

### ERG examination
After 16 h dark adaptation, mice were anesthetized followed by pupil dilation at 7 days post-treatment. Lubricant eye gel was used to maintain corneal clarity. One electrode was placed in direct contact with the cornea. The reference and ground electrodes were placed subcutaneously in the back neck and tail, respectively. ERGs were recorded using a Ganzfeld scotopic ERG system (Phoenix Research Labs, USA) under 0.1 log cd*s/m² flash intensity.

### qRT-PCR of Bax and caspase-3
Eyes were enucleated at 3 days post-treatment and retinas were collected. Total RNA of retinas was extracted using a universal RNA purification kit (EZB-RN4, EZBioscience, USA). Then mRNAs were reversed into cDNA

using a color reverse transcription kit (EZB-A0010CGQ, EZBioscience, USA) according to the manufacturer's protocol. RT-PCR was carried out in 384-well plates containing primers, cDNA, and EZB-A0012-R1 (EZBioscience, USA) by ABI 7900 fast. mRNA relative expression levels were calculated by the $2^{-\Delta\Delta Ct}$ method, with GAPDH as the internal control. Primers of GAPDH, Bax, and caspase-3 are shown in Supplementary Table 1.

### Western blotting
Eyes were enucleated at 3 days post-treatment. Proteins were extracted from retinas, and the concentrations were measured using BCA methods. The protein samples were subjected to SDS-polyacrylamide gel electrophoresis and transferred to a polyvinylidene fluoride (PVDF) membrane (Millipore, USA). The blots were incubated with primary antibody for Bax (ab32503, 1:2000, abcam, USA) and caspase-3 (ab184787, 1:2000, abcam, USA) at 4 °C overnight, following incubation with HRP-conjugated goat anti-rabbit IgG Ab (7074 S, Cell Signal Technology, USA) at room temperature for 1 h. The protein bands were visualized using an ECL Chemiluminescence kit (E422, Vazyme, China).

### RGC-5 cell model construction and apoptosis analysis
RGC-5 cells were cultured in DMEM medium, containing 10% FBS and 1% penicillin/streptomycin at 37 °C in a 5% $CO_2$ incubator. 400 μM NMDA was added to the culture medium to induce RGC damage. PBS, MSC-sEVs, miR22-sEVs, or con-sEVs were added respectively afterwards. After 72 h, cell viability was evaluated using a CCK8 assay according to the manufacturer's instructions. Apoptosis was assessed by flow cytometry, with Annexin V/PI double staining. The protein expression of Bax and caspase-3 was detected by western blotting. Apoptosis activity was tested by pre-treating the cells with 50 μM Z-VAD-FMK (IZ0050, Solarbio, China) 1 h before injury model construction. The protein expression of cleaved PARP was detected by western blotting (44-698 G, 1:1000, anti-cleaved PARP antibody, Thermo Fisher Scientific, USA).

### Flow cytometry analysis
Cells were diluted with the binding buffer into a concentration of $10^6$ cells/mL. Annexin V and propidium iodide (PI) were added to the sample and incubated for 20 min at room temperature. After cells were resuspended in PBS, flow cytometric analysis was performed with a BD FACSCalibur. Data were analyzed using flowjo_v10 software (FlowJo, LLC, USA). Gating strategy was shown in Supplementary Fig. 1.

### Target gene prediction
MiRNA targets were identified using publicly available bioinformatics tools (Targetscan, miRDB, and DIANA). DAVID (http://mirtarbase.mbc.nctu.edu.tw/) bioinformatics tool was applied to perform enrichment analyses of these target genes. The mRNA expression of *Max, Arrb1, Csf1r, Map3k12, Akt3* and *Tgfbr1* were measured by qRT-PCR. Primers are shown in Supplementary Table 1.

### Dual luciferase reporter assay
TargetScan database was used to predict the potential binding sites for miR22 in the *Map3k12* mRNA 3'-UTR. The wild-type (WT) sequence of *Map3k12*-3'-UTR (WT 3'-UTR) and mutated sequence of *Map3k12*-3'-UTR (mutant 3'-UTR) were subcloned into the upstream of the luciferase reporter gene. The dual luciferase reporter gene plasmids were co-transfected into 293 T cells with miR22 mimics. After 24 h, dual-luciferase reporter assay was performed using a Dual-Luciferase Reporter Assay System (E1910, Promega, USA).

### Pathway analysis of miR22
After NMDA-induced RGC-5 damage model was established, miR22 mimics or miR22 inhibitors (GenePharma, Suzhou, China) were trans-fected using lipofectamine 3000 (Life Technologies, USA). The protein expression of MAP3K12, ERK, p38 and caspase-3 were detected by western blotting (anti-MAP3K12 antibody, MAP3K12-101AP, 1:500, Thermo Fisher Scientific, USA; anti-ERK antibody, 4695 S, 1:1000, Cell Signaling Technology, USA; anti-p38 antibody, 9212 S, 1:1000, Cell Signaling Technology, USA).

### Statistics and reproducibility
Each experiment was repeated at least three times. Differences between two groups were compared using an unpaired two-tailed Student's *t*-test. One-way analysis of variance (ANOVA) with Bonferroni post-hoc comparisons was used for the analysis of three or more groups. All data were analyzed by using GraphPad Prism10 (GraphPad Software, Inc., La Jolla, CA92037, USA). *$P < 0.05$ was considered significant for each analysis.

### Reporting summary
Further information on research design is available in the Nature Portfolio Reporting Summary linked to this article.

### Data availability
All data supporting the findings of this study are available within the paper or in the Supplementary Information file. Numerical source data can be found in Supplementary Data 1 and original western blots in Supplementary Fig. 2. All other data are available from the corresponding author upon a reasonable request.

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

## Acknowledgements

X.Z. discloses support for the research of this work from National Natural Science Foundation of China [grand number 82371044], Tianjin Health Commission Key Discipline Special Project [grand number TJWJ2023XK009], and Tianjin Key Medical Discipline (Specialty) Construction Project [grand number TJYXZDXK-037A]. B.Y. discloses support for the research of this work from National Natural Science Foundation of China [grand number 81800825], Open Project of Tianjin Key Laboratory of Retinal Functions and Diseases [grand number 2021tjswmq002], Tianjin Medical University "Clinical Talent Training 123 Climbing Plan", and Tianjin Medical University Eye Hospital High-level Innovative Talent Program [grand number YDYYRCXM-E2023-01]. We would like to thank Editage for English language editing.

## Author contributions

B.Y.: financial support, collection and assembly of data, manuscript writing. K.W.: data analysis and interpretation, collection and assembly of data. H.H.: data analysis and interpretation, collection and assembly of data, manuscript revision. Y.L.: data analysis and interpretation, manuscript writing. Y.Y.: collection and assembly of data, manuscript writing. X.L.: provision of study material, administrative support. X.X.: conception and design, provision of study material, final approval of manuscript. X.Z.: conception and design, final approval of manuscript, manuscript revision. All authors have read and agreed to the published version of the manuscript.

## Competing interests

The authors declare no competing interests.
