## [Peer review file · Communications Biology]

Reviewers' comments:

Reviewer #1 (Remarks to the Author):

In this manuscript, Bo Yu et al investigate the impact of microRNA-22-3p (miR22) enriched extracellular vesicles derived from mesenchymal stem cells (MSC-Exos) on N-methyl-D-aspartic acid (NMDA)-induced retinal ganglion cell (RGC) injuries.

The authors report that miR22-Exos exhibited neuroprotective properties against NMDA-induced RGC injuries. Indeed, treatments with miR22-Exos resulted in thicker retina layers, reduced apoptosis, preserved RGCs, and seem to improve retinal functions in vivo. In vitro assays suggested that miR22-Exos promoted RGC viability that the authors attempted to link to the mitogen-activated protein kinase kinase kinase 12.

Here are major points that need to be addressed to clarify the experimental data and support the authors' conclusions.

1- Cell Death: The level of expression of Caspase-3 (RNA and protein) is a poor marker for cell death, as it does not measure its activity per se. The authors should evaluate Caspase-3 activity with western-blot for cleaved Caspase-3 and PARP substrate, together with a Caspase-3 activity assay (many are commercially available), to support their claims in Figures 3B – 4C – 6D. Other markers need to be examined in Figure 3A (Fas, p53, Bcl-2, Bax, caspase-8...). Please also check in Figure 4B whether z-vad or qvd might reverse the cell death effects. In Figure 4, the authors describe that EVs impact proliferative capability of the RGC-5 cell line using CCK8 assay and AnnexinV/PI flow cytometry. These tests are usually used to determine cell viability and cell death, rather than proliferation rate. Did the authors perform EdU/BrdU assay to confirm their conclusions?

2- EV characterization: The authors need to provide a more meticulous characterization of their EVs (please refer to MISEV guidelines). First, the term 'exosome' should be avoided given the lack of specificity of their methods. Negative markers should be used for Figure 1C to control for cellular contaminants (any organelle markers). It is also required that the authors test for the possible presence of Argonaute and apolipoproteins in their EV preparations. The authors need to assess for the EV number and size distribution in the different conditions shown in Figure 1B. Likewise, the authors need to perform RNase protection assay.

3- miR22-EV versus miR22 mimics: as it is, the paper appears divided in evaluating either EV-delivered miR22 or miR22. It is important to compare both approaches throughout the manuscript. In Figures 2A, the authors should include miR22 mimic as a control. Conversely, in Figures 5C and 6C-D, the author should include EV-delivered miR22.

4- In vivo models: The authors should document the intraocular pressure to control for potential ocular side effects associated with repeated intravitreal injections. Histological analysis of retinal tissue from treated animals has to be improved to clarify their initial data and quantify any changes in RGC density, retinal thickness, fibrosis and inflammation infiltrates. Animals treated with miR-22-3p-overexpressing EVs could be evaluated at multiple time points post-treatment to assess the sustained preservation of retinal ganglion cells and retinal function. This would provide valuable insights into the long-term efficacy of the treatment.

5- Mechanistic insights: The link between miR22-Exos, Map3k12 and cell viability have to be consolidated. Did the authors investigate miR22-Exos effect on the expression of other effectors (JNK, p38, ERK) of this pathway? What would be the effect of Map3k12 siRNA on RGC-5 viability treated as in Figure 4. Cell cycle assay should also be performed to examine whether NMDA-induced RGC injury and sEVs-miR22 impact on cell cycle (given the role of Map3k12).

6- Information: Some figures are hard to read, as crucial information is missing from legends, making the interpretation more difficult. In the Method section, the authors often refer to previously described protocols, without given much details, not even verified sources. Please revise thoroughly.

7- TNF: In Figure 3, the authors report a decrease in the TNF mRNA expression level upon sEVs-miR22 treatment. The impact of sEVs-miR22 is not further investigated/mentioned. This should be either removed from the manuscript or studied more in-depth the inflammatory pathway.

Reviewer #2 (Remarks to the Author):

Yu et al explore the use of mesenchymal stem cell derived extracellular vesicles on retinal ganglion cell injury. Specifically, MSCs were transfected with miR-22. EVs containing miR-22 were then used for mouse and in vitro experiments and showed improvement of markers of RGC injury over MSC-EV treatment alone. miR-22 target analysis was performed and one gene in the MAPK pathway was shown to be a target of miR-22 in vitro. The study is timely and on an important target. I have some comments that would improve the manuscript.

The title needs some changes. Overexpressed should be Overexpressing. Also, the data doesn't support that it is through regulating MAPK pathway. This pathway was one that was altered but the mechanism was not shown. Tone down this part of the title.

Because the biogenesis pathway for exosomes can not be confirmed the preferred terminology in the field is to use "extracellular vesicles". Please see Witwer and Thery JEV, 2019 and the new MISEV 2023 (Welsh et al., JEV 2024)

Line 76 The MSC characterization data should be shown. There should be a little bit more information at the beginning of the results about what type of MSC (source etc) and what was being transfected. An important control would also be to show that lentivirus transfection does not change the characteristics of MSCs.

For EV characterization it is optimal according to MISEV guidelines to examine another EV marker, from a different category (ex. Flotillin etc) and a negative marker (ex. GM130 etc)

Line 90 treatment timeline is not clear here. After injury when do you treat with EVs and how long after that do you stain? It is just mentioned how long after injury that TUNEL staining was performed. I see it is in the methods but would be nice to add to the results for clarity.

Line 113 "proved" is too strong of a word here.

Figure 3 different labels are used in the figure than in the rest of the manuscript. More appropriate actually since EVs is used.

Figure 4A proliferation is not necessarily examined here just # of cells or viability. So references to proliferation should be avoided.

Figure 5 do levels of miR-22 change in recipient cells of miR-22-Exos? If conclusions want to be drawn that this is indeed the mechanism than this should be shown.

Line 254 how were MSCs purified from human umbilical cord blood? Details and references should be included.

Line 275 were EVs collected after 48 hours or what time frame?

Line 281 "previous protocols" is indicated but is missing references. Also describe in more detail the processing for EM imaging.

Line 290 how were these concentrations for injection derived? Were 1 ul of each added as well?

Line 364 details for luciferase assay should be included, was a kit used, if so which one?

Another point is that human MSCs were used in the mouse model? This should be discussed.

Response to reviewers:

Comments from Reviewer 1

Reviewer #1 (Remarks to the Author):

In this manuscript, Bo Yu et al investigate the impact of microRNA-22-3p (miR22) enriched extracellular vesicles derived from mesenchymal stem cells (MSC-Exos) on N-methyl-D-aspartic acid (NMDA)-induced retinal ganglion cell (RGC) injuries.

The authors report that miR22-Exos exhibited neuroprotective properties against NMDA-induced RGC injuries. Indeed, treatments with miR22-Exos resulted in thicker retina layers, reduced apoptosis, preserved RGCs, and seem to improve retinal functions in vivo. In vitro assays suggested that miR22-Exos promoted RGC viability that the authors attempted to link to the mitogen-activated protein kinase kinase kinase 12.

Response: The referee shows a thorough understanding of the manuscript, and we thank him for his careful review. We have addressed each of the points below. The improvements will surely aid the reader's understanding, while the adding of experiments will strengthen the quality of the manuscript.

Here are major points that need to be addressed to clarify the experimental data and support the authors' conclusions.

1- Cell Death: The level of expression of Caspase-3 (RNA and protein) is a poor marker for cell death, as it does not measure its activity per se. The authors should evaluate Caspase-3 activity with western-blot for cleaved Caspase-3 and PARP substrate, together with a Caspase-3 activity assay (many are commercially available), to support their claims in Figures 3B – 4C – 6D. Other markers need to be examined in Figure 3A (Fas, p53, Bcl-2, Bax, caspase-8...). Please also check in Figure 4B whether z-vad or qvd might reverse the cell death effects. In Figure 4, the authors describe that EVs impact proliferative capability of the RGC-5 cell line using CCK8 assay and AnnexinV/PI flow cytometry. These tests are usually used to determine cell viability and cell death, rather than proliferation rate. Did the authors perform EdU/BrdU assay to confirm their conclusions?

Response 1: Thanks for your suggestion very much. As your suggestion, we measured another apoptosis-related cytokine (Bax) in RGC-5 cell line and the RGC injury animal model. The results have been added in Figure 3 and 4. We also tried to do western-blot for cleaved caspase-3 for many times. However, the blots were not good, which is shown below (Reply Figure 1). Thus not been added in the manuscript. As to the CCK8 assay in Figure 4. Thank you for your kindly reminding. Sorry for the misused word here. We have revised the term "proliferation" to "viability".

Reply Figure 2: Western-blot for cleaved caspase-3. The blots can't be shown clearly.

2 - EV characterization: The authors need to provide a more meticulous characterization of their EVs (please refer to MISEV guidelines). First, the term 'exosome' should be avoided given the lack of specificity of their methods. Negative markers should be used for Figure 1C to control for cellular contaminants (any organelle markers). It is also required that the authors test for the possible presence of Argonaute and apolipoproteins in their EV preparations. The authors need to assess for the EV number and size distribution in the different conditions shown in Figure 1B. Likewise, the authors need to perform RNase protection assay.

Response 2: Thanks for your suggestion very much. First, as your suggestion, after reading the article "the new MISEV 2023 (Welsh et al., JEV 2024)", we revised the inappropriate term "exosomes" to "small extracellular vesicles (sEVs)" because the diameter of the vesicles we used in the current research were less than 200nm. We also added some experiments to further characterize sEVs. Western blotting was used to detect protein expressions of GM130 (an organelle marker), β -actin (a cell marker) and TSG101 (another type of EV marker) in cells and sEVs (Fig. 1D). NanoSight NS300 (Malvern, UK) was used to measure the size distribution of sEVs (Fig. 1C).

3- miR22-EV versus mir22 mimics: as it is, the paper appears divided in evaluating either EV-delivered miR22 or miR22. It is important to compare both approaches throughout the manuscript. In Figures 2A, the authors should include miR22 mimic as a control. Conversely, in Figures 5C and 6C-D, the author should include EV-delivered miR22.

Response 3: Thanks for your suggestion very much. I agree with it. Here are some reasons that we design the study like this. It has been long realized that natural miRNA efficacy is normally limited for therapeutic purpose as they are easily degraded by RNases and/or they could stimulate the innate immune system through activating Toll-like receptors (PMID: 35806173, 15723075). The effect of miR22 is not the focus of the research. We've also referred to some similarly designed literatures (PMID: 35255957, 34134765). The first part of the current study is to explore the therapeutic effect of MSC-sEVs and the enhanced effect of miR22-modified MSC-sEVs. The second part is to explore possible mechanisms of the enhanced effect. Our conclusion "miR22-sEVs ameliorate NMDA-induced RGC injury through the inhibition of MAPK signaling pathway" can be drawn by the current design. In future studies, we will pay more attention to the

study design and make it more reasonable. Thank you again for your professional suggestion.

4 - In vivo models: The authors should document the intraocular pressure to control for potential ocular side effects associated with repeated intravitreal injections. Histological analysis of retinal tissue from treated animals has to be improve to clarify their initial data and quantify any changes in RGC density, retinal thickness, fibrosis and inflammation infiltrates. Animals treated with miR-22-3p-overexpressing EVs could be evaluated at multiple time points post-treatment to assess the sustained preservation of retinal ganglion cells and retinal function. This would provide valuable insights into the long-term efficacy of the treatment.

Response 4: The IOP increased temporally while intravitreal injection, which can be reflected by the transient corneal edema during injection. We measured IOP of each mouse by icare an hour later. The IOP was normal then. But we ignored to stress this procedure in the current draft. Thus, we add a sentence "IOP was measured an hour later to ensure the security of intravitreal injection" to describe it (highlighted in the method part). We only did intravitreal injection once as described in the draft "For treatment groups, 1 μ l PBS, 0.5mg/mL MSC-Exos, 0.5mg/mL miR22-Exos, or 0.5mg/mL exosomes derived from control scramble lentivirus transfected MSCs (con-Exos) were intravitreally injected simultaneously with NMDA". Maybe it sounds a little confusing. As your suggestion, we revised it to "For the PBS group, 1 μ l NMDA (20mM) + 1 μ l PBS was intravitreally injected to the right eye of each mouse. For the other three treatment groups, 1 μ l NMDA (20mM) + 1 μ l MSC-sEVs (0.5mg/mL), 1 μ l NMDA (20mM) + 1 μ l con-sEVs (0.5mg/mL) or 1 μ l NMDA (20mM) + 1 μ l miR22-sEVs (0.5mg/mL) was intravitreally injected to the right eyes of mice in each group individually".

HE staining results of day 7 were added in Fig. 2, and were described in the result part (highlighted). Because cells in ganglion cell layers (GCL) are not only RGC. RGC density can be better tested by wholemount than HE staining. Retinal thickness can be measured by OCT. We didn't do similar statistical analysis in HE staining sections. HE staining were done after 3 days, 7 days and 14 days of treatment in pre-experiments. The results were shown below. We found the retinal structures of 14 days post-treatment were similar with that of 7 days post-treatment (Reply Figure 2). So, we decide to measure other retinal structural and functional parameters in the time point of 7 days after treatment in the current article. The focus of the research was short-term effect of sEVs. The first limitation of the current study is that we only studied the short-term therapeutic effects of MSC-sEVs and miR22-sEVs, which has been listed in the end of the discussion part. As your suggestion, we also added the word "short-term" in abstract and discussion of the manuscript to make the current conclusion more reasonable. As Reply Figure 2 shows, the beneficial effect can also be seen after 14 days, providing relative long-term efficacy (Not shown in this manuscript). Hope for your understanding.

Reply Figure 2: Representative HE staining pictures at different time points post-injury. At 3 days post-injury, the retinal thickness were similar in all groups. Cells in GCL shrink in PBS group. Retinal thickness decreased after 7 days and 14 days of NMDA injury. Retinas in three sEVs-treated groups were thicker than that of the PBS group. Cell numbers in GCLs were larger in three treatment groups at 7 days and 14 days post-injury.

5- Mechanistic insights: The link between miR22-Exos, Map3k12 and cell viability have to be consolidated. Did the authors investigate miR22-Exos effects on the expression of others effectors (JNK, p38, ERK) of this pathway? What would be the effect of Map3k12 siRNA on RGC-5 viability treated as in Figure 4. Cell cycle assay should also be performed to examine whether NMDA-induced RGC injury and sEVs-miR22 impact on cell cycle (given the role of Map3k12).

Response 5: Thank you for your suggestion. As your suggestion, we detect other effectors of MAPK pathway. What's more, we also test the effect of miR22 inhibitor on the pathway. The protein expression levels of MAP3K12, ERK, P38 and caspase-3 were tested after miR22 mimics or inhibitors were added. The purpose of Fig 4 is to show the therapeutic effect of different sEVs in the NMDA-induced RGC-5 injury model, which is consistent with the in vivo results in Fig 2 and Fig 3. The following experiments were done to explore the enhanced effect of miR22, which were shown in Fig 5 and Fig 6. This is the whole design of the study. The consideration you proposed is the regions we want to deeply explored in future study. Now some pre-experiments were done. We found MAP3K12 siRNA decreased the viability of RGC-5 in the NMDA-induced cell injury model. The results were shown blow (Reply Figure 3). This is the beginning of our next paper. We didn't put it in the current study. Hope for your understanding. In future study, we will focus on the impact of miR22 or miR22-sEVs on cell cycles, and further explain the exact mechanisms of the influence on apoptosis. We appreciate your kindly suggestion very much.

Reply Figure 3: MAP3K12 siRNA on RGC-5 viability. After NMDA treatment, the RGC-5 viability decreased. MAP3K12 siRNA increased RGC-5 viability significantly (**** $P < 0.0001$).

6 - Information: Some figures are hard to read, as crucial information is missing from legends, making the interpretation more difficult. In the Method section, the authors often refer to previously described protocols, without given much details, not even verified sources. Please revise thoroughly.

Response 6: We have revised figure legends one by one. The references were also added in the methods part to explain previously described protocols as your suggestion (References 19, 41, 42). All revisions were highlighted in the text.

7 - TNF: In Figure 3, the authors report a decrease in the TNF mRNA expression level upon sEVs-miR22 treatment. The impact of sEVs-miR22 is not further investigated/mentioned. This should be either removed from the manuscript or studied more in-depth the inflammatory pathway.

Response 7: Thank you for your suggestion. We only focus on apoptosis not inflammation. Thus, we have removed the TNF- α result from Fig. 3, Fig. 4 and relative parts.

Comments from Reviewer 2

Reviewer #2 (Remarks to the Author):

Yu et al explore the use of mesenchymal stem cell derived extracellular vesicles on retinal ganglion cell injury. Specifically, MSCs were transfected with miR-22. EVs containing miR-22 were then used for mouse and in vitro experiments and showed improvement of markers of RGC injury over MSC-EV treatment alone. miR-22 target analysis was performed and one gene in the MAPK pathway was shown to be a target of miR-22 in vitro. The study is timely and on an important target. I have some comments that would improve the manuscript.

Response: We kindly thank Reviewer for the constructive criticism and helpful suggestions. We have addressed each of the points below. The improvements should help the readers better understand how the study fits within the broader areas of research and strengthen the manuscript overall.

1. The title needs some changes. Overexpressed should be Overexpressing. Also, the data doesn't support that it is through regulating MAPK pathway. This pathway was one that was altered but the mechanism was not shown. Tone down this part of the title.

Response 1: Thank you for your suggestion. We have changed the word "Overexpressed" to "Overexpressing". We also added some experiments to detect the protein expression of the effectors (ERK, P38) in MAPK pathway. We found their protein expression levels changed together with MAP3K12 and cleaved caspase-3 after miR22 mimics or inhibitor added (shown in Fig.6). Based on the current results, we preliminarily think the effect was through regulating MAPK pathway.

2. Because the biogenesis pathway for exosomes cannot be confirmed the preferred terminology in the field is to use "extracellular vesicles". Please see Witwer and Thery JEV, 2019 and the new MISEV 2023 (Welsh et al., JEV 2024)

Response 2: As your suggestion, after reading the article "the new MISEV 2023 (Welsh et al., JEV 2024)", we revised the inappropriate term "exosomes" to "small extracellular vesicles (sEVs)" because the diameter of the vesicles we used in the current research were less than 200nm.

3. Line 76 The MSC characterization data should be shown. There should be a little bit more information at the beginning of the results about what type of MSC (source etc) and what was being transfected. An important control would also be to show that lentivirus transfection does not change the characteristics of MSCs.

Response 3: MSCs we used in the current research were the same with one of our previously published articles (The therapeutic effect and mechanism study of small extracellular vesicles (sEVs) derived from mesenchymal stem cells (MSCs) on retinal light injury. [Chinese journal of experimental ophthalmology]). MSC characterization was shown in that paper as blow (Reply Figure 4). We didn't show repeated results here. We've referred to several articles and found that microRNAs lentivirus transfection doesn't change the characteristics of MSCs (PMID: 28042326, 30224076, 29178928). Thank you

for your kindly remind. We know some of the references were lack after the sentence “as previously reported”. We have added the references (highlighted References 19, 41, 42). We also added more details about the MSC source and transfection procedures as your suggestion in culture of MSC part.

Reply Figure 4: Identification of MSCs and MSC-sEVs A: Flow cytometry showed 99.8% MSCs were CD90-positive B: Flow cytometry showed 99.5% MSCs were CD105-positive C: Flow cytometry showed very few MSCs were CD34-positive D: Flow cytometry showed few MSCs were CD45-positive E: Passage 1 MSCs showed spindle-shaped and adherent under a light microscope ($\times 4$, bar=500 μ m) F: MSC-sEVs showed a circular vesicle structure with a diameter of 80-140 nm by transmission electron microscopy ($\times 12\,000$, bar=100 nm)

4. For EV characterization it is optimal according to MISEV guidelines to examine another EV marker, from a different category (ex. Flotillin etc) and a negative marker (ex. GM130 etc)

Response 4: For EV characterization, we’ve added some experiments. Western blotting was used to detect protein expressions of GM130 (an organelle marker), β -actin (a cell marker) and TSG101 (another type of EV marker) in cells and sEVs. NanoSight NS300 (Malvern, UK) was used to measure the size distribution of sEVs. The data were shown in Fig.1 C and 1D.

5. Line 90 treatment timeline is not clear here. After injury when do you treat with EVs and how long after that do you stain? It is just mentioned how long after injury that TUNEL staining was performed. I see it is in the methods but would be nice to add to the results for clarity.

Response 5: To make the timeline clearer. We added some sentences in the methods part (highlighted) to stress the time points for each treatment or measurement.

6. Line 113 “proved” is too strong of a word here.

Response 6: We’ve changed the word “proved” to “indicated” to make it sound more reasonable.

7. Figure 3 different labels are used in the figure than in the rest of the manuscript. More appropriate actually since EVs is used.

Response 7: We have deleted all the “exosomes”. The final term we use is “sEVs” in the whole manuscript and all figures.

8. Figure 4A proliferation is not necessarily examined here just # of cells or viability. So references to proliferation should be avoided.

Response 8: Thank you for your suggestion. CCK8 only detect viability of the cells. We didn't do proliferation assays. We have revised the term “proliferation” to “viability”.

9. Figure 5 do levels of miR-22 change in recipient cells of miR-22-Exos? If conclusions want to be drawn that this is indeed the mechanism than this should be shown.

Response 9: Thank you for your suggestion. Sorry about the negligence of showing these data in the current version. We added a statistical chart in Figure 5 (Fig. 5 C).

10. Line 254 how were MSCs purified from human umbilical cord blood? Details and references should be included.

Response 10: MSCs were derived from human umbilical cord, not umbilical cord blood. Sorry for the unclear expression of the MSC isolation and culture procedures. We have added details and references in this part as your suggestion as follows. “MSCs were cultured and identified as previously described. Briefly, Human umbilical cords were washed and cut into pieces in PBS, and then were sequentially digested with collagenase II and trypsinization at 37 °C.” (highlighted)

11. Line 275 were EVs collected after 48 hours or what time frame?

Response 11: The culture medium was collected from P3-P5 MSCs 48 hours after passaging. We revised the sentence “The cultured supernatants were collected and subjected to gradient centrifugation as previously described” to “The cultured supernatants were collected 48 hours after passaging and subjected to gradient centrifugation as previously described”.

12. Line 281 “previous protocols” is indicated but is missing references. Also describe in more detail the processing for EM imaging.

Response 12: We have added a reference here (Reference 42) and also the details of EM imaging procedure (highlighted) as follows. “10µl sEVs suspension was placed onto Formvar-carbon coated copper grids. The grid was then moved to a solution of phosphotungstic acid (50µl, pH 7.0) for 5 minutes. After air-drying, the sample was examined under an electron microscope at 80 kV.”

13. Line 290 how were these concentrations for injection derived? Were 1 ul of each added as well?

Response 13: Selection of treatment concentrations was first based on our previous research. In our previously published article (PMID: 27686625), the therapeutic effect of MSC-sEVs at different concentrations were tested in a mouse retinal laser injury model.

0.5mg/mL was shown to be the most effective. In the current pre-experiments, we used 0.1mg/mL, 0.5mg/mL or 1mg/mL MSC-sEVs to treat the mice and test the retinal function by ERG 7 days post treatment. We found 0.5mg/mL and 1mg/mL MSC-sEVs showed similar a and b wave amplitudes. Finally, we used the concentration of 0.5mg/mL in the current study. 1µl was added in each treatment group. To avoid confusion, we've revised the sentence to "For the PBS group, 1µl NMDA (20mM) + 1µl PBS was intravitreally injected to the right eye of each mouse. For the other three treatment groups, 1µl NMDA (20mM) + 1µl MSC-sEVs (0.5mg/mL), 1µl NMDA (20mM) + 1µl con-sEVs (0.5mg/mL) or 1µl NMDA (20mM) + 1µl miR22-sEVs (0.5mg/mL) was intravitreally injected to the right eyes of mice in each group individually" (highlighted).

14. Line 364 details for luciferase assay should be included, was a kit used, if so which one?

Response14: The Dual-Luciferase Reporter Assay System (Promega, USA) was the kit used in this assay. As your suggestion, we've added the kit name in the manuscript. Thank you again for your reminding.

15. Another point is that human MSCs were used in the mouse model? This should be discussed.

Response 15: The use of human MSCs were due to the consideration of its possible clinical use in future. Many of the MSC-related published articles use human MSCs as their therapeutic strategy (PMID: 36096356, 37303049, 35842714, 35477552). Our designs are similar with them. The references (Ref. 20-23) we cited in the second paragraph of the discussion part were all about the use of human MSCs in the treatment of RGC injury animal models. Thus, we stressed the source of MSCs in related part and added some sentences to explain this (highlighted). We appreciate your suggestion very much.

Reviewers' comments:

Reviewer #1 (Remarks to the Author):

The authors have made some progress in addressing the concerns raised during the initial review. However, several crucial experiments and controls are still required.

1. Cell Death: The revised data are still not sufficiently convincing, particularly regarding the correlation between expression and activity. While the authors addressed the issue with Bax, other suggested markers have not been tested. Please also note that Bax is not a cytokine as stated in the rebuttal. Moreover, additional experiments are needed to support apoptosis data, including PARP cleavage analysis, utilization of a pan-caspase activity kit, and investigation into the effects of zQVD or zVAD, as outlined in the first review.
2. Extracellular Vesicle (EV) Characterization: Although improvements have been made in this section, there is still a need to evaluate the potential contamination with Argonaute and Apolipoproteins.
3. Rationale for Experimentation: The justification provided for not attempting the suggested experiments remains insufficient. The authors should either conduct the recommended experiments or extensively discuss the current limitations.
4. Acceptable Revision: This aspect of the revision meets the required standards.
5. Inclusion of sEV Data: In my opinion, data on small extracellular vesicles (sEVs) are pertinent to the current study and should be incorporated.

Reviewer #2 (Remarks to the Author):

The authors have addressed my concerns and the manuscript is significantly improved.

Response to reviewers:

The authors have made some progress in addressing the concerns raised during the initial review. However, several crucial experiments and controls are still required.

Response: Thank you and we appreciate for your serious scientific attitude and time spent on reviewing the current paper. We have addressed each of the points below. We look forward that the adding of experiments will strengthen the quality of the manuscript and finally get your approval.

1. Cell Death: The revised data are still not sufficiently convincing, particularly regarding the correlation between expression and activity. While the authors addressed the issue with Bax, other suggested markers have not been tested. Please also note that Bax is not a cytokine as stated in the rebuttal. Moreover, additional experiments are needed to support apoptosis data, including PARP cleavage analysis, utilization of a pan-caspase activity kit, and investigation into the effects of zQVD or zVAD, as outlined in the first review.

Response 1: Thank you for your suggestion. We have added the experiments as suggested to better prove the correlation between expression and activity of apoptosis. We performed the apoptosis activity analysis experiments, which included western blotting analysis for cleaved PARP protein expression, and investigation of the zVAD effect in RGC injury cell models. The results were added in Fig. 4 (Fig. 4E).

2. Extracellular Vesicle (EV) Characterization: Although improvements have been made in this section, there is still a need to evaluate the potential contamination with Argonaute and Apolipoproteins.

Response 2: Thank you for your suggestion. When we characterize EVs extracted from serum or other body fluids, we will evaluate the potential contamination of apolipoproteins by examining protein expression of APOA1. However, for EVs derived from MSC, we think the contamination of Argonaute and Apolipoproteins will be very rear. We also referred to other similar articles (PMID: 31973741, 32127037, 31857259). So, we examined different types of positive expressed proteins and one negative expressed protein for EV characterization. We are willing to make improvements step by step forward.

3. Rationale for Experimentation: The justification provided for not attempting the suggested experiments remains insufficient. The authors should either conduct the recommended experiments or extensively discuss the current limitations.

Response 3: We agree with your opinion. We have revised the limitation of the study in the discussion part (highlighted in yellow) to extensively discuss the current limitations.

4. Acceptable Revision: This aspect of the revision meets the required standards.

Response 4: Thank you for your approval on the previous modifications.

5. Inclusion of sEV Data: In my opinion, data on small extracellular vesicles (sEVs) are pertinent to the current study and should be incorporated.

Response 5: Thank you for your kindly suggestion very much. As we have mentioned in initial response letter, the current study can be divided into 2 parts. The first aim was to test the beneficial effect of miR22-sEVs. The second aim was to explore possible reasons for the enhanced beneficial effect of adding miR22. Thus, in Fig.5, we did the bioinformatics analysis for the target gene prediction of miR22 first. If we set four groups (adding the sEV group) in the following target gene verification experiments, the outcomes may appear somewhat perplexing and convoluted. Therefore, in future research, we hope to explore the sEVs' effect on apoptosis deeply and intactly as per your suggestion, which will help us to understand the thorough mechanisms. Nevertheless, the current research will be a foundation for the future studies and researchers to explore the role of small extracellular vesicles (sEVs). We have elaborated in the discussion section (highlighted in yellow).